# A Bayesian Statistical Model Is Able to Predict Target-by-Target Selection Behaviour in a Human Foraging Task

**Alasdair D. F. Clarke** [1,*] , **Amelia R. Hunt** [2] **and Anna E. Hughes** [1]

1   Department of Psychology, University of Essex, Wivenhoe Park, Colchester CO4 3SQ, UK
2   School of Psychology, University of Aberdeen, King's College, Aberdeen AB24 3FX, UK
*   Correspondence: a.clarke@essex.ac.uk

**Abstract:** Foraging refers to search involving multiple targets or multiple types of targets, and as a model task has a long history in animal behaviour and human cognition research. Foraging behaviour is usually operationalized using summary statistics, such as average distance covered during target collection (the path length) and the frequency of switching between target types. We recently introduced an alternative approach, which is to model each instance of target selection as random selection without replacement. Our model produces estimates of a set of foraging biases, such as a bias to select closer targets or targets of a particular category. Here we apply this model to predict individual target selection events. We add a new start position bias to the model, and generate foraging paths using the parameters estimated from individual participants' pre-existing data. The model predicts which target the participant will select next with a range of accuracy from 43% to 69% across participants (chance is 11%). The model therefore explains a substantial proportion of foraging behaviour in this paradigm. The situations where the model makes errors reveal useful information to guide future research on those aspects of foraging that we have not yet explained.

**Keywords:** foraging; visual search; bayesian model; decision; strategy

## 1. Introduction

Foraging, the act of searching for and gathering multiple targets (such as food), has been studied in both nonhuman animal contexts [1] and in human psychology studies [2]. Foraging engages a wide range of different perceptual, cognitive, decision-related and motor skills, and is an ecologically relevant behaviour for most species, including humans. For these and other reasons, a sustained interest in foraging has developed our understanding of how we search for multiple instances of multiple types of targets. For example, the classic marginal value theorem [3] is generally good at predicting when an organism will decide to stop searching in a patch and move on to another, based on the finding rate dropping below an expected rate, and taking into account the relative energy costs of traveling between patches versus staying within a patch. There has also been a concerted effort to characterize the spatial patterns of foraging behaviour. The Lévy Walk, for example, has been argued to be a good description of the foraging path many species take when the locations of targets are unknown (e.g., [4], but see also [5]).

Another general principle of foraging is that of the "search image", which describes the perceptual features that a foraging animal can use to identify targets or classes of targets [6]. A limited capacity for complex search images means that dividing attention over more than one kind of potential search target can impede search, particularly when what distinguished targets from distractors is not a simple feature like colour or motion. As a result, foragers tend to search in "runs" of one type of target before switching to another. In humans, this behaviour has been studied using computer-based displays of targets and distractors (e.g., [2]), in which participants must "collect" targets (by clicking or tapping on them) and ignore distractors. When targets can be identified based on a single,

easily distinguishable feature (for example, find all the red and green shapes and ignore the blue and yellow shapes) participants tend to switch frequently between different types of targets, taking a relatively efficient path through the array. In contrast, when targets are defined based on a combination of more than one feature (e.g., find all the red squares and green circles and ignore the red circles and green squares), participants tend to switch far less frequently, often selecting all the targets of one category and then all the targets of the other. Consequently, the path taken to collect the targets is less efficient. In other words, frequent switching between target types limits the distance travelled between targets, while collecting all the targets of each type before switching to the other type sacrifices some efficiency of movement in service of reducing the mental workload.

Different aspects of foraging behaviour, such as the search image and the search path, clearly interact with one another, but most research has tried to understand them separately. In many cases, the pattern of foraging behaviour has been studied using aggregate measures calculated on the level of a trial, such as the mean number of 'runs' (where the same type of target is selected multiple times in a row) or the total number of targets found during the longest run. These summary statistics can provide a broad view of switching behaviour, but they can also be biased by the spatial distribution of target types, and by a forager's preference for one target or another. To address the need for a more precise set of measurements, we recently developed a new model to analyse foraging data, based on a sampling without replacement procedure [7]. By using this model, we demonstrated that we were able to break down foraging into a number of different cognitive biases, such as a preference to stick to the same target type, or a preference for nearby targets, and used this model to successfully reanalyse data from a number of open access datasets [2,8–10].

While our model is able to do a good job of allowing us to understand biases at the level of a trial (e.g., conjunction vs. feature search differences [2] or the influence of high value targets compared to low value targets [11]), we did not originally investigate to what extent our model was able to predict target-by-target behaviour within any given trial. However, as the model is based on target-by-target level information, it is possible for us to use this information to ask: to what extent does our model predict foraging behaviour within a trial? Can it predict exactly which target a participant will pick next? If the model makes mistakes, can we understand where it is failing, in order to improve our understanding of how human foraging behaviour operates?

The original implementation of our model has little to say about the initial target selection in each trial: as there is not yet a previous target, the stick/switch and proximity parameters are all ignored leaving just a simple salience parameter that allows us to model one set of targets as being more attractive than the other. Taking inspiration from the work on the central viewing bias in the eye movements literature [12,13] we develop two models of the spatial bias in initial target selection during visual foraging.

## 2. Materials and Methods

### 2.1. Datasets

We use a previously published dataset to explore how well our model can account for trial-level behaviour: the visual foraging data from Clarke and colleagues [8]. This is an attractive dataset for our needs as it is very close to the "classic" visual foraging paradigm conducted by Kristjánsson and colleagues [2], yet with a substantially larger sample of participants. (The key difference between paradigms is that [2] used an ipad and finger foraging, while [8] used a desktop computer and mouse clicks.) Furthermore, we have already demonstrated that our model can capture individual differences in foraging behaviour [8]. As such, we will only give a brief overview of the data here and refer the reader to these earlier papers for more details.

### 2.2. A Model for Visual Foraging

We will make use of the foraging model from Clarke et al. (2022) [7]. This treats foraging as a sampling without replacement process in which each item $i$ has probability $p_i$ of being selected as the next target. The $p_i$ depend on four parameters:

- $b_A$—preference for selecting items of type $A$ rather than $B$.
- $b_S$—preference for selecting items of the same type as the previously selected item.
- $\sigma_d$—preference for selecting items close to the previously selected item.
- $\sigma_\theta$—preference to keep selecting items along a straight line versus changing direction.

These combine to give:

$$w_i = g(b_a t_i + b_s m(t_i, t_{i-1})) \times \rho_d(i, i-1)\rho_\theta(i, i-1) \tag{1}$$

where $t_i = 1$ if item $i$ is of class $A$ and 0 otherwise, and $m(t_i, t_{i-1}) = 1$ if item $i$ is the same class as the previously selected item. $\rho_d$ and $\rho_\theta$ measure proximity and the effect of direction:

$$\rho_d = e^{-\sigma_d d(i,j)} \tag{2}$$

$$\rho_\theta = e^{-\sigma_d \theta(i,j)} \tag{3}$$

$d(i, j)$ is simply the Euclidean distance between items $i$ and $j$, while $\theta$ is defined as:

$$\theta(i, j) = \frac{f(\text{atan2}(i, j) - \text{atan2}(i-1, i))}{\pi} \tag{4}$$

with $f(\phi_1, \phi_2) = \min((\phi_1 - \phi_2)\%2\pi, (\phi_2 - \phi_1)\%2\pi)$ calculating the angular difference. atan2 is the direction of travel from $i$ to $j$. This model is implemented in a multilevel framework, allowing each of the four parameters to vary from participant to participant. Further details including priors and full code can be found with [7].

Note: while our model returns estimates of full posterior probability distribution for each parameter, to reduce computational complexity and make it easier to compare to the empirical data, we will work with the means of these distributions for our parameter values.

### 2.3. Software Environment

We mainly used R v4.2.0 and rStan v2.26.11 [14] (R Foundation for Statistical Computing, Vienna, Austria). For fitting the model to the data from [8] we used a university computing cluster with R v3.6.3 and rStan v2.19.2.

## 3. Results: Evaluating the Model

To assess how well our model can account for behaviour at the level of individual trials, we start by stepping through each trial in the data from Clarke and colleagues [8]. For each target selection (Note: for the reasons discussed above, we ignore the initial target selection for now, as the model has very little to say about it. We return to this issue in Section 4), in each trial, we can estimate how likely each of the remaining items are to be selected using the parameters from our posterior model. As can be seen in Figure 1, the model is putting somewhere between 25% and 75% of the weight on the target that is selected next, easily outperforming a chance $1/n$ baseline. There are also interesting differences between the two conditions. For example, in the conjunction condition, there is a clear 'jump' around target 20, which probably reflects the tendency for participants to forage in runs, selecting all exemplars of one target and then all exemplars of the other: there is a decrease in accuracy as people switch and then an increase again. We can also see that the model is well calibrated in that the probabilities assigned to the most likely target manage to capture how often that target is actually selected. However, we can see a decrease in accuracy when it comes to how often "runner-up" candidate items are selected as our model appears to systematically undervalue these (presumably by putting too much of the probabilistic weight on low chance items).

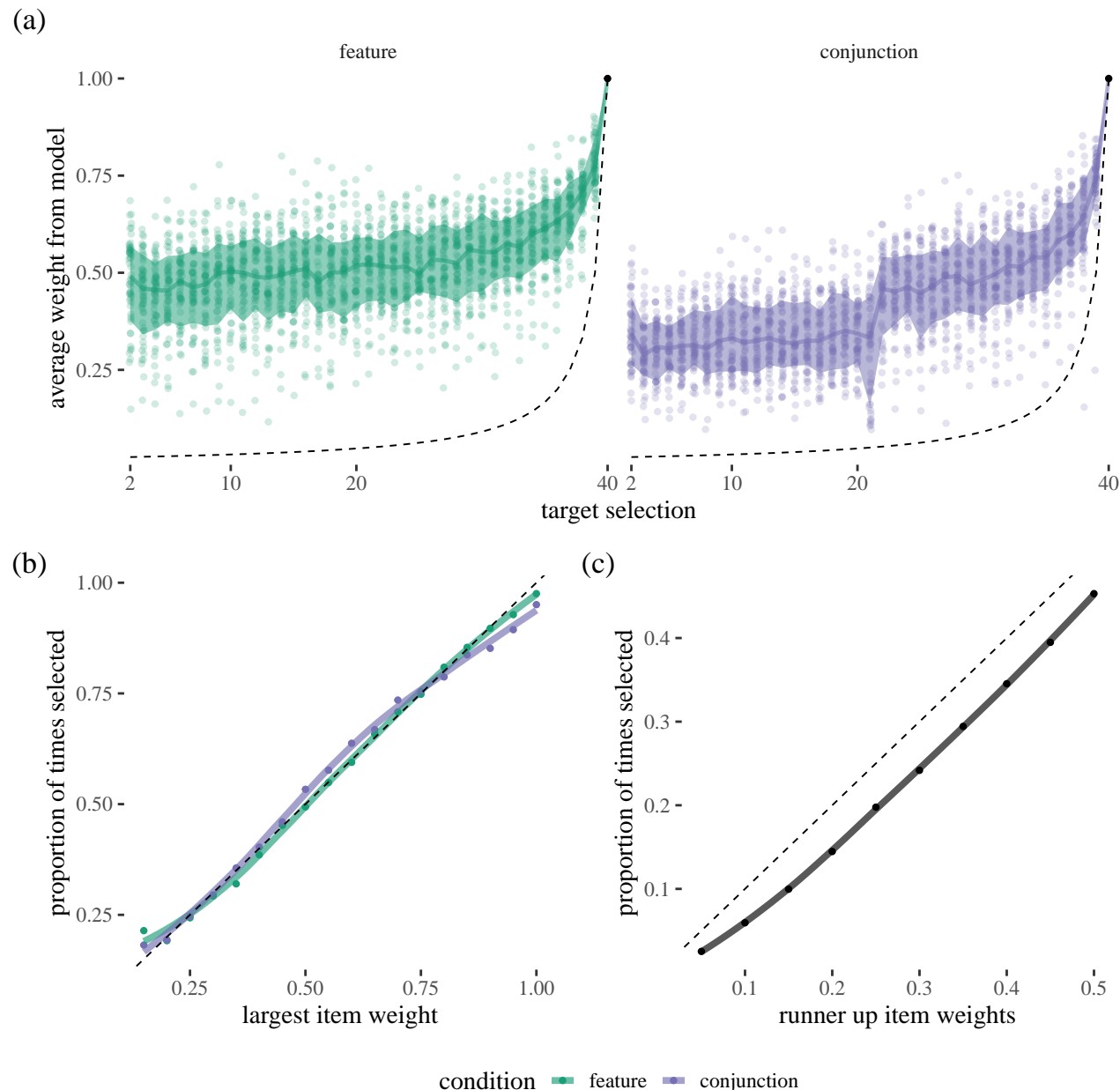

**Figure 1.** (**a**) Top left: Posterior probabilities for target selections during a visual foraging task. Each dot shows the data from an individual participant, averaged over trials, in the feature condition and the shaded region indicates the interval in which we expect 67% of participants to fall. The dashed line indicates chance performance. Top right: As top left, except showing the conjunction condition. (**b**) Calibration plot for our foraging model. The *x*-axis gives the largest weight assigned by the model while the *y*-axis shows how often that target was actually selected by a human participant. The dashed line here and in the right plot is the identity line. (**c**) This plot shows how often the model selects the 2nd and 3rd ranked items based on the weights assigned by the model.

We summarise accuracy for each person by calculating the mean proportion of times the model assigns the most weight to the item that was then selected for each participant (This metric isn't perfect as it averages over all target selections within a trial, and these clearly have different baseline probabilities: 2.5–100%. However, it has the upside of being more intuitive than anything else we could think of). These accuracy scores can be compared to our uniform chance baseline that gives a value of 10.9% over the course of

a trial. Figure 2 hints at some individual differences when it comes to how predictable the model is: we can see a lot of variation in the size of the weights. However, interestingly, we find that this variability can be explained by our model parameters: nearly all of the differences between different participants, as well as the within-participant differences in the feature and conjunction conditions, can be explained by the *bP* parameter, which is our proximity bias. The model does a better job of predicting which will be selected next when there is a stronger proximity bias (see Supplementary Materials: Part 1 for more details).

We now look at some example trials and compare the behaviour of our human participants to the predicted behaviour from our model (see Figure 3). When looking at trials in which our model has done a particularly good job of accounting for the target selections (model accuracy > 80%) we can see that the *disagreements* often occur in cases when the human participant appears to be carrying out some form of local path length optimisation. In general, the cases in which our model suggests a different target from the one the participant actually selected appear reasonable.

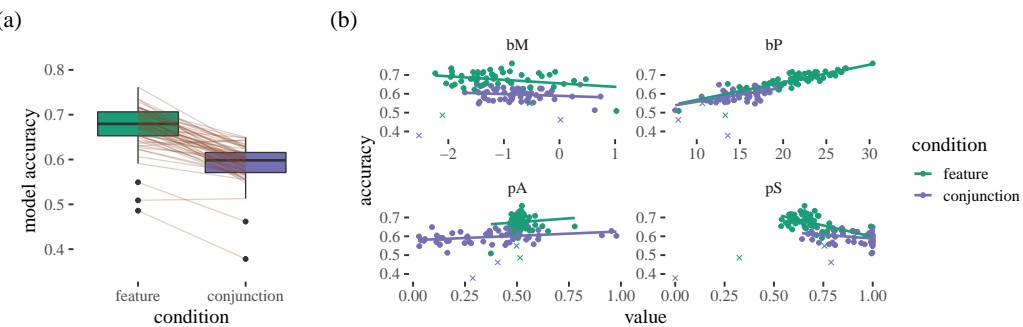

**Figure 2.** (**a**) Prediction scores for participants. Boxplots show quartile range and the grey lines indicate individual participants. The dots indicate outliers. (**b**) How accuracy of our model varies with the strength of an individual's *bM*, *bP*, *pA* and *pS* parameters. We can see two clear outlier participants (marked with an X).

For trials in which the model does a worse job of predicting (accuracy < 50%) we can see that the human behaviour appears less organised in terms of proximity in general and it seems unlikely that adding some form of path-length optimisation to our model would improve accuracy with these participants. We can see this more clearly in Figure 4: while our model does a good job of capturing the average total path length, there is considerable variation with some participants generating shorter paths and some longer. We can see that this appears to be systematically related to how well our model can predict behaviour. Interestingly, in some cases, poorly predicted trials appear to have very strong *pS* biases, collecting all exemplars of one target and then the other: thus, the participants are carrying out a strategy that the model is not capturing well. This fits with conjunction trials being harder for the model to predict, as these trials generally have fewer switches.

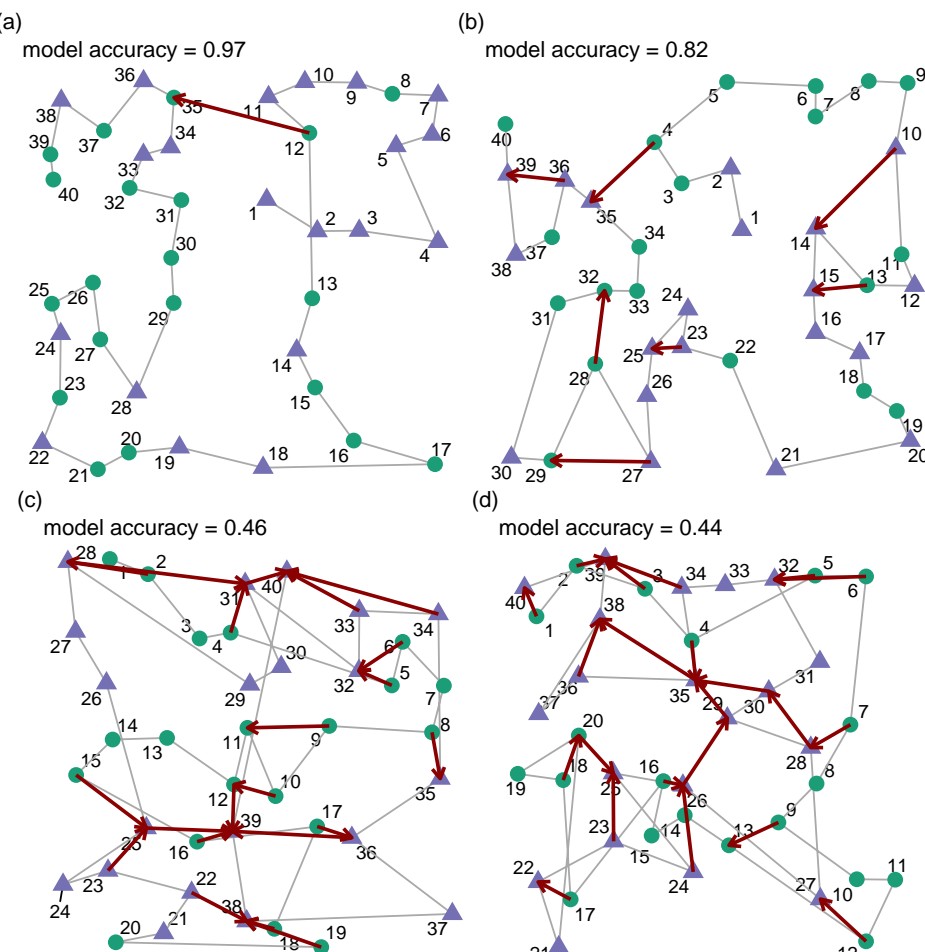

**Figure 3.** (**a**,**b**) Two randomly selected trials in which the model does a good job in accounting for human behaviour. The numbers indicate the order the participant selected targets in. Red arrows indicate places where the model prediction deviates from participant behaviour. When participants diverge from the model's prediction, it appears to be due to some form of path-length optimisation. (**c**,**d**) Two randomly selected examples in which the model does a less good job in accounting for the order in which human participants selected the items.

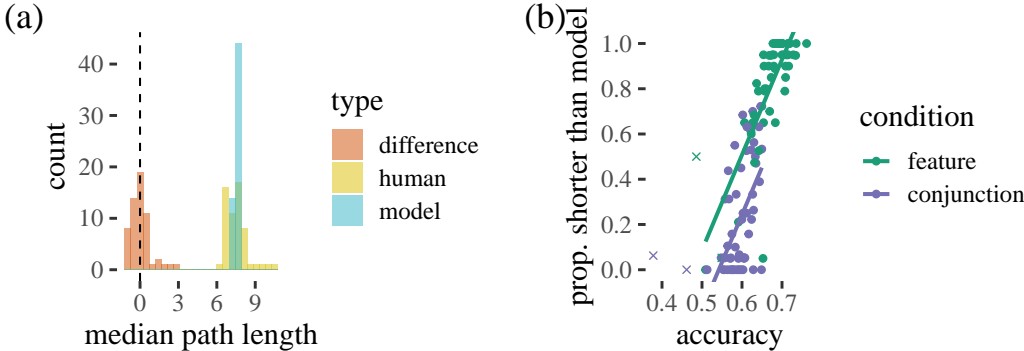

**Figure 4.** (**a**) Histogram showing the distribution of the median path lengths for our human participants and model (fitted to each human participant via the random effect structure). (**b**) The relationship between model accuracy and the proportion of trials in which the human participant has a shorter total path length than predicted.

#### 4. Improving the Model: Location of the First Target Selection

One weakness of the foraging model by Clarke et al. (2022) [7] is that it makes little attempt to predict which item will be selected first. In this section we aim to improve this by modelling participant bias (or preference) in their choice of initial target selection. Our approach is inspired by previous work on the central bias in scene viewing [12,13]. These studies fit truncated Gaussian distributions to the $(x, y)$ coordinates of fixation locations. Here we will do similar, and fit beta distributions (We use beta distributions here as it is difficult to specify a mixture model using truncated Gaussians in Stan. However, beta distributions may well be a more appropriate choice) to locations of the initial target selections in the visual foraging task.

Figure 5 shows the distribution of the locations of first target selections. We can see that unlike the central bias in fixations during scene viewing, the distribution of initial target locations appear to be bimodal: while most targets are positioned in the top left hand corner, there is a second smaller distribution of central target selections. This appears to be due to different participants choosing to utilise different strategies rather than within-subject variation. However, we have no evidence that these different initial strategies affect the model parameters in any systematic way (see Supplementary Materials: Part 1). We try two different modelling approaches (multilevel and mixture) to assess whether the less complex mixture model is able to account for the patterns of starting positions seen.

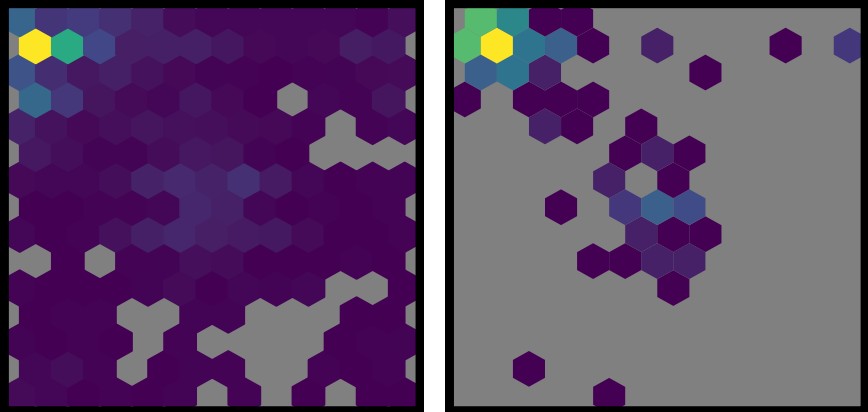

**Figure 5.** Hexagonal heatmaps showing the two-dimensional distribution of the location of initial target selections. Grey areas indicate cells with a count of 0. The left panel shows the distribution over all initial target selections (i.e., multiple trials per participant) while the right panel shows the distribution of each participant's median initial selection (i.e., each participant contributes one data point to the graph).

#### 4.1. Multi-Level Modelling Results

We first consider a multi-level model in which we fit beta distributions to the $x$ and $y$ coordinates of each participant's initial target selections (Full details of the analysis can be found in the Supplementary Materials: Part 1):

$$x_i \sim Beta(a_{x,i}, b_{x,i}) \tag{5}$$

$$y_i \sim Beta(a_{y,i}, b_{y,i}) \tag{6}$$

The results are shown in Figure 6 and we can clearly see that most participants either have a strong bias to $x = 0$ and $y = 0$, or a more diffuse central bias.

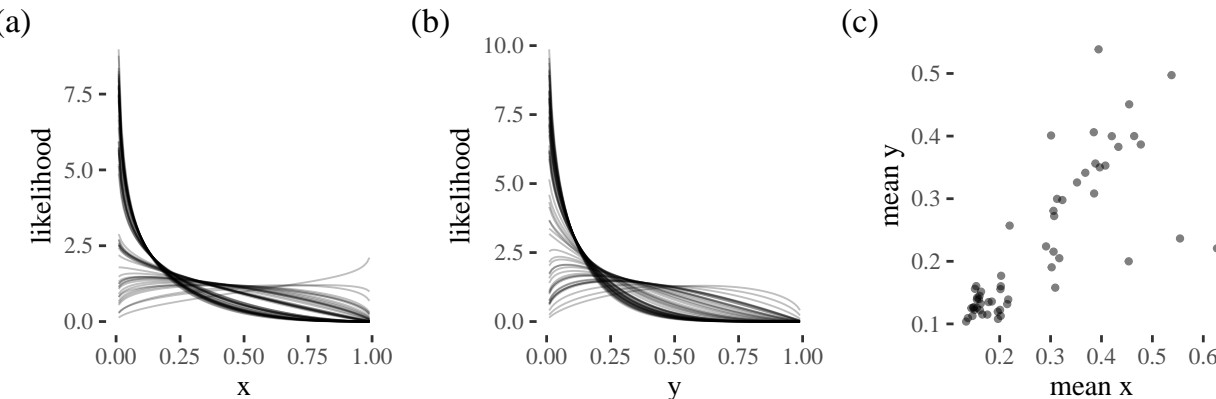

**Figure 6.** Posterior fits for (**a**) horizontal and (**b**) vertical locations of the initial target selection. Each line represents a different participant. (**c**) A scatter plot showing the *x* and *y* posterior means for each participant.

### 4.2. Mixture Modelling

While the multi-level model outlined above appears to offer a good fit for the between-subject differences in initial target selection location, it comes at the expense of requiring four parameters per participant (=290 for the 58 participants in our dataset). Given that the majority of our participants appear to be following one of two distinct strategies, we can potentially simplify our model by using a two-component mixture model:

$$x_i \sim \lambda_i \times Beta(a_{x,1}, b_{x,1}) + (1 - \lambda_i) \times Beta(a_{x,2}, b_{x,2}) \tag{7}$$

$$y_i \sim \lambda_i \times Beta(a_{y,1}, b_{y,1}) + (1 - \lambda_i) \times Beta(a_{y,2}, b_{y,2}) \tag{8}$$

This reduces the number of parameters to eight (to specify the two Beta distributions) and then one $\lambda$ value per participant (=66 in our dataset). While the formulae in Equations (7) and (8) appear more complex than those in Equations (5) and (6), the model requires less than a quarter as many free parameters.

The model fit is shown in Figure 7. We can see that the two components identified by the model clearly correspond to the top corner and diffuse-central strategies discussed above. We can also see from the lambda values that our participants take on a range of different mixtures between the two.

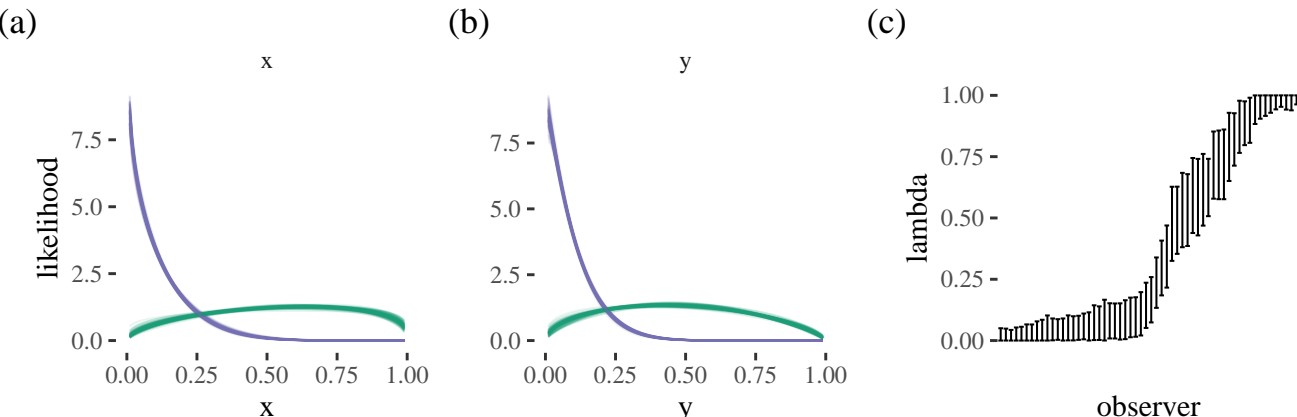

**Figure 7.** Posterior fits for the (**a**) horizontal and (**b**) vertical locations of the initial target selection. The two colours represent the two components in the mixture model. (**c**) shows the mixing parameters for each participant, with the error bar indicating the 95% HDPI.

### 4.3. Posterior Predictions

To compare the two approaches, we assess how well they fit the empirical data: we can use the fitted distributions to calculate the weight assigned to the selected target by each method for each trial in our dataset (Figure 8). We can see that both models give a similar distribution of target weights and accuracies. Overall, both methods select the correct target in around half of the trials. Interestingly, as above, there is considerable variation between participants, with participants who favour the top-left corner being easier to predict.

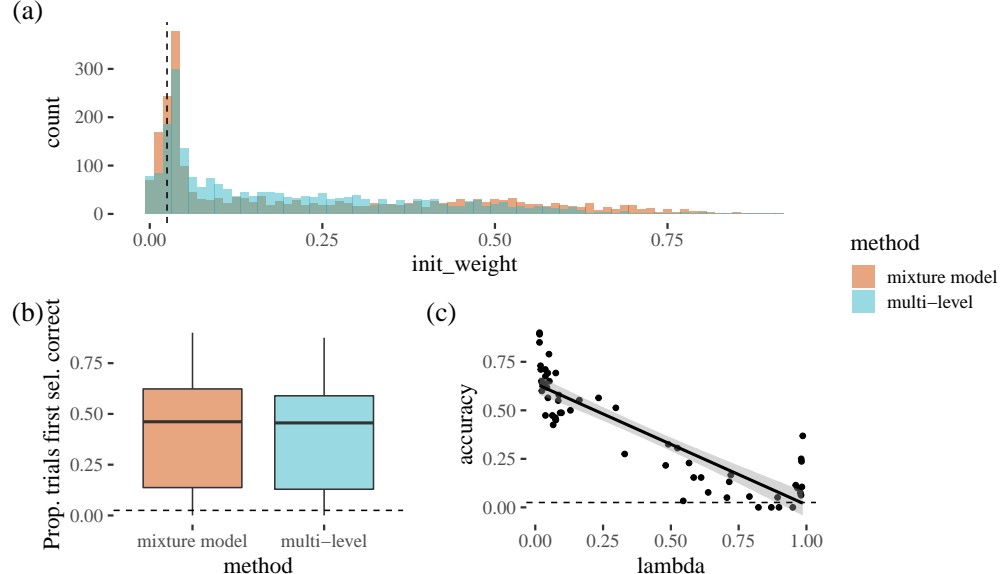

**Figure 8.** (**a**): a count of initial weight values predicted from both methods (mixture and multi-level). (**b**): the proportion of times the first selection was correct for both methods. (**c**): as lambda (the model parameter underlying initial weight) increases, accuracy increases i.e., the model is more accurate for the cases where participants are selecting a corner as their initial target selection. The dots each represent an individual participant, and the line is the line of best fit.

### 4.4. Replication

To test how well our methods generalise, we used the dataset from [2]. We split this into a training dataset (50% of the original data i.e., 10 trials per participant) and a test dataset (the remaining 50% of the data). The foraging model is fit on the training dataset, and the parameters from this model are then used to predict behaviour at the level of individual trials. We show that our methods appear to generalise beyond [8]: in particular, feature foraging was more predictable than conjunction foraging, and the model is well calibrated, with participants more frequently selecting targets with higher model assigned weights. Proximity also seems to be an important factor in determining model accuracy, at least for the feature condition, although the relatively low number of participants in this experiment makes it harder to draw strong conclusions. One interesting difference compared to [8] is in initial target selection: while there was a strong bias towards starting in a corner, there was little evidence of central bias, perhaps because this experiment was completed on an iPad. Further details and graphs can be found in the Supplementary Materials: Part 2.

### 5. Discussion

In our original model of visual foraging [7], we were able to robustly measure aggregate parameters that underlie behaviour, at both an overall mean level across an experiment, and at the level of individual participants. In the current manuscript, we asked a more difficult question: could this model make target-by-target predictions, guessing which target would be picked next within a trial? We found that for some trials our model was

over 80% accurate, demonstrating that at least in some cases, our model is able to make good predictions of target behaviour.

A benefit of our approach is that it is relatively easy to interrogate the model to try to understand why prediction accuracy is higher for some participants and trials than others. The model is generally better at feature search compared to conjunction search, and also seems to be better at predicting participants with a stronger proximity bias. The model also seems to assign more weight to 'runner-up' items than real humans do. Real foragers thus seem to seriously consider only a few nearby targets (although they do still have some low probability of selecting a further away target), in contrast to the model, in which the item weights assigned fall off more gradually with distance.

However, we can also delve deeper into exploratory hypotheses about other factors that might influence predictability. One suggestion is that the people who our model is able to predict are in some way the most 'optimal'. If this is the case, we might expect that model predictability should correlate with how good people are at the task. There is some evidence that on a trial-by-trial basis, shorter trials are easier for the model to predict, and on a target-by-target basis, faster moves from one target to the next are more easily predicted by the model (see Supplementary Materials: Part 1). This likely is linked to the fact that our model is better at predicting participants with a stronger proximity bias, as targets that are closer to the previous target are also likely to be selected more quickly. Intuitively, it makes sense that the model makes poorer predictions when a participant has exhausted a local patch and may need to make a bigger jump to a more distant area to carry on collecting targets, as there may be multiple candidates and distance may no longer be such a good predictor (for example, people may choose to move to the next most densely populated patch).

There has been some previous work on within-trial behaviour in the context of foraging. For example, ref. [11] found that intertarget times vary across a trial (with targets selected later in the trial being slower) and intertarget distances also vary (with targets selected later in the trial having greater intertarget distances). Similarly, ref. [10] showed that switches between target categories can be characterised by a change in foraging 'tempo', and if participants are forced to forage at a particular speed, by asking them to synchronise with an auditory signal, higher tempos led to a systematic decrease in the probability of participants switching between target categories. In the current version of the model, we do not consider time explicitly (although distance is likely to be correlated with time in this task). However, it would be possible to extend the model to incorporate timing information, and this may help to improve the target-by-target predictions.

From inspection of Figure 3, we noticed that in some cases, participants may be using a type of local path minimisation procedure, selecting a further away target in order to minimise overall path length, which the model does not always seem to predict. Could this help to explain some of the mistakes the model makes? Overall, the model does not seem to make substantially longer paths through the items than the human participants. However, there is a relationship between model accuracy and the proportion of times that participants select paths shorter than the model: as model accuracy increases, people seem to be more likely to have paths shorter than the model predicts. This seems to suggest that path minimisation is not a good explanation for the participants our model is failing to capture, and instead the people that are difficult to predict are taking longer paths than the model thinks they should take.

Our exploratory analyses seem to hint at the idea that the people who are difficult for the model to predict are those who do not behave in a manner that optimises path length. It could be that these participants are simply unpredictable, and do not behave in a consistent manner on a target-by-target basis, making it difficult for a model to predict their behaviour at this level of granularity. However, it is possible that there are other factors that we have not taken into account that could further help explain these discrepancies. Participants who 'stick' to one type of target are likely to have longer path lengths, and it seems that the model does do a poorer job of predicting conjunction trials, which generally have stronger

*pS* biases. Thus, one area for future improvement of the model is to try to capture this behaviour better: it may be that a 'floor' parameter (on a participant-by-participant basis) in the spatial fall off, allocating more weight to more distant targets, may help account for currently difficult to predict trial behaviour.

There are also many other factors that the model does not account for. For example, it is possible that relatively unpredictable participants use multiple rules and switch between them in ways that we are not currently captured in the model. Some participants may also lose track of where they are more frequently (perhaps due to attentional lapses) and thus may show foraging patterns that are predictable apart from unpredictable discontinuities, which again may not be easy to model. Another possibility is that participants may make exploratory, information-seeking visits to different parts of the array due to uncertainty about the targets present: this may be particularly important for conjunction trials, where peripheral visual information about shape is likely to be poorer. Finally, the displays had artificial discontinuities (i.e., screen edges) which may lead to different behaviour in different participants: some may carry out the task as if they are reading, reaching the right hand edge and then swinging back to the left, whereas others may 'bounce' off an edge they have just reached. Our model may well be able to predict the latter, but make less good predictions for the former behaviour.

To a large extent, initial item selection strategy seems to be stable within a participant, with people either starting in the corner or the centre of the screen. Our model is therefore also relatively accurate for selecting the initial target, selecting the correct target in around half of the trials (with a mixture model performing similarly to a multilevel model with many more parameters). Participants who prefer the top left corner were easiest to predict, probably because these participants formed the majority group in this experiment. Our results are congruent with [15], who found a bottom left bias in a 3D environment. They suggest this may be an advantageous strategy because it allows your foraging to be more organised (e.g., in an 'S-shape', as has been found in real world search tasks [16]). The smaller group of participants who prefer to start in the centre of the screen may be displaying similar behaviour to the well-known central bias in eye movements [12,17,18] However, it is worth noting that these findings probably depend strongly on the context of the task. For example, there was no obvious central bias subgroup in [2], possibly reflecting the fact that this experiment was carried out on an iPad, compared to the computer screen and mouse used in [8]. One challenge for future modelling work is to what extent we should try to account for these types of task-specific details: by incorporating them, we can generate better predictions for the experiment we are currently modelling, but perhaps at the cost of generalisability to other paradigms.

One of the key benefits of computational models is that they allow us to rigorously test how well we are able to predict behaviour, and can provide insights into what factors we are not capturing that may have important influences on how participants complete a task. We have demonstrated that our foraging model [7] is not only able to predict behaviour in aggregate, but can also make reasonable predictions at the target-by-target level. It is particularly good if a participant has a strong proximity bias, and on 'feature' trials where participants are normally switching between target types fairly frequently. It is also good at predicting where participants will start on a trial. The model finds it more difficult to predict 'discontinuous' jumps, which could be caused by a range of factors: a more difficult foraging task (e.g., conjunction searches), local path length minimisation, inattention, or aspects of the physical search space (e.g., edges of the screen). However, it would be possible to extend the model to incorporate these factors e.g., by incorporating a heuristic that would allow for local path minimisation, or introducing time as a predictor in the model. Our findings also suggest possible future directions for empirical work, such as evaluating the effect of inattention on foraging behaviour, or how the foraging targets are organised in space. We suggest that computational modelling is a powerful tool for helping us to understand behaviour, both by incorporating previous research into a shared framework, and by making testable predictions for future work.

**Supplementary Materials:** The following are available online at https://www.mdpi.com/article/10
.3390/vision6040066/s1, Document S1: Supplementary Materials Part 1, Document S2: Supplementary Materials Part 2.

**Author Contributions:** Conceptualization, A.D.F.C., A.R.H. and A.E.H.; methodology, A.D.F.C.;
software, A.D.F.C.; formal analysis, A.D.F.C.; data curation, A.D.F.C. and A.E.H.; writing—original
draft preparation, A.D.F.C.; writing—review and editing, A.R.H. and A.E.H.; visualization, A.D.F.C.
and A.E.H.; funding acquisition, A.D.F.C. and A.R.H. All authors have read and agreed to the
published version of the manuscript.

**Funding:** This research was funded by the Economic and Social Research Council grant number
ES/S016120/1 to A.D.F.C. and A.R.H.

**Institutional Review Board Statement:** Not applicable (secondary data analysis only).

**Informed Consent Statement:** Not applicable (secondary data analysis only).

**Data Availability Statement:** Data supporting reported results can be found on Github at https:
//github.com/scienceanna/foraging_svg, accessed on 25 April 2022. The datasets analysed in this
study can be found at https://osf.io/y6qbv/, accessed on 25 April 2022, and https://journals.plos.
org/plosone/article?id=10.1371/journal.pone.0100752, accessed on 27 May 2022.

**Acknowledgments:** The authors would like to thank all the researchers who publicly shared
their data.

**Conflicts of Interest:** The authors declare no conflict of interest.

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
