# Peer review of "A Bayesian Statistical Model Is Able to Predict Target-by-Target Selection Behaviour in a Human Foraging Task"

_2411-5150, 2014_

Round 1

Reviewer 1 Report

This is a well-written and interesting paper that is suitable for publication in Vision. I only have minor concerns. First, two broad strokes comments:

Foraging is described as engaging a lot of systems, yet no reference is made to action or the motor system, which is the most fundamental of systems foraging engages isn't it?

I wonder if some of the inaccuracy is due to the absence of modeling uncertainty in the visual array. Of course, we enjoy higher resolution / finer detail within foveal and central vision, and the eye movements are pre-planned based on a combination of information in the periphery and perhaps working or brief (short term) memory. The model presumably has access to a perfect representation of "what is out there", does it not? And so I wonder if the model could be improved by adding some form of information seeking component to the selection of the targets. Perhaps subjects, for example, violate a nearest-neighbour heuristic in order to acquire a 'snapshot' of some other part of the array, reloading working memory, so to speak, for planning the next batch of targets to select/saccades to make. Perhaps this is what the authors are hinting at when they write in the discussion " ...'peakier' choice distribution ..."? Then again, perhaps the iPad screen size is too small to tax visual working memory in any serious way. In any case, I don't expect the authors to create a new model for this paper. Just food for thought.

Introduction and Methods are tightly written and appropriate, imo.

In the results, what is meant by the "largest target" on line 123? Is size a feature of the targets in the foraging task? Or does large refer to something else?

Certain aspects of Figure 1 I found a little ambiguous. I can understand why the authors wanted to merge the two conditions together in the leftmost and middlemost panels. Nevertheless, I think the leftmost panel would look clearer if the two data for the two conditions were separated. If they were the top two panels in a four-panel figure, they would be fairly easy to compare even when separated. This is a minor concern, and mostly stylistic, although it does look like there is some overlap in the data between conditions, which it what prompted my suggestion to separate them to improve the display [For comparison, in the right panel of Figure 2, the density of the data points is less and so the overlap seems OK.]. For the right panel, the caption refers to items being selected, but I am guessing it is referring to the model based selection. If that's true, then I would suggest adding something like "the model" to "...how often [the model selects] the 2nd and 3rd ranked items..."

line 137 "account" should be "accounting"

line 138 consider revising or deleting "appear to", as it suggests the authors aren't sure. Also, the same sentence uses "appears" later on.

line 142: "...we do a worse job of predicting..." stick with using "the model" rather than saying "we" when it comes to referring to the predictions of the model. As a more general comment, I would like to see the model personified less. So, for example, it's described as "struggling" on line 127. I admit this is a stylistic issue, and so the authors should enjoy leeway. I guess I just prefer a more dispassionate, less personal tone.

Figure 5 is displaying the subject data, correct? Either way, it would be good to make this a little clearer in the caption.

Discussion is also tightly written and seems appropriate to me.

Author Response

This is a well-written and interesting paper that is suitable for publication in Vision. I only have minor concerns. First, two broad strokes comments:

Foraging is described as engaging a lot of systems, yet no reference is made to action or the motor system, which is the most fundamental of systems foraging engages isn't it?

Thank you for pointing this out, we completely agree and we have now updated the introduction accordingly (L19).

I wonder if some of the inaccuracy is due to the absence of modeling uncertainty in the visual array. Of course, we enjoy higher resolution / finer detail within foveal and central vision, and the eye movements are pre-planned based on a combination of information in the periphery and perhaps working or brief (short term) memory. The model presumably has access to a perfect representation of "what is out there", does it not? And so I wonder if the model could be improved by adding some form of information seeking component to the selection of the targets. Perhaps subjects, for example, violate a nearest-neighbour heuristic in order to acquire a 'snapshot' of some other part of the array, reloading working memory, so to speak, for planning the next batch of targets to select/saccades to make. Perhaps this is what the authors are hinting at when they write in the discussion " ...'peakier' choice distribution ..."? Then again, perhaps the iPad screen size is too small to tax visual working memory in any serious way. In any case, I don't expect the authors to create a new model for this paper. Just food for thought.

Thank you for your thoughts here - yes, the model has a “perfect” representation and we have not tried to account for e.g. central/peripheral perceptual differences. As you say, this would be a very interesting direction for future work, and could certainly explain some of our results here, not least because the feature trials in the dataset we are modelling are defined by colour (which should be easy to discriminate in the periphery) whereas the conjunctions trials are defined by colour and shape (and thus may be less easy to discriminate in the periphery, requiring more exploration). We have added a suggestion to this effect to the Discussion (L267).

Introduction and Methods are tightly written and appropriate, imo.

In the results, what is meant by the "largest target" on line 123? Is size a feature of the targets in the foraging task? Or does large refer to something else?

Apologies, this was sloppy wording on our part - we meant “most likely”. This has been fixed (L117).

Certain aspects of Figure 1 I found a little ambiguous. I can understand why the authors wanted to merge the two conditions together in the leftmost and middlemost panels. Nevertheless, I think the leftmost panel would look clearer if the two data for the two conditions were separated. If they were the top two panels in a four-panel figure, they would be fairly easy to compare even when separated. This is a minor concern, and mostly stylistic, although it does look like there is some overlap in the data between conditions, which it what prompted my suggestion to separate them to improve the display [For comparison, in the right panel of Figure 2, the density of the data points is less and so the overlap seems OK.]. For the right panel, the caption refers to items being selected, but I am guessing it is referring to the model based selection. If that's true, then I would suggest adding something like "the model" to "...how often [the model selects] the 2nd and 3rd ranked items..."

We have modified Figure 1 and clarified the figure caption as suggested.

line 137 "account" should be "accounting"

Fixed (L135).

line 138 consider revising or deleting "appear to", as it suggests the authors aren't sure. Also, the same sentence uses "appears" later on.

Deleted, as suggested (L136).

line 142: "...we do a worse job of predicting..." stick with using "the model" rather than saying "we" when it comes to referring to the predictions of the model. As a more general comment, I would like to see the model personified less. So, for example, it's described as "struggling" on line 127. I admit this is a stylistic issue, and so the authors should enjoy leeway. I guess I just prefer a more dispassionate, less personal tone.

Thank you for your helpful comments here - we have fixed the examples suggested (e.g. L140, L274) and have tried to make the language a little more formal throughout the manuscript.

Figure 5 is displaying the subject data, correct? Either way, it would be good to make this a little clearer in the caption.

The left hand part of Figure 5 is displaying all initial target selections (i.e. multiple trials per participant) and the right hand side is a median value for each participant (i.e. each participant contributes one data point to the graph). We have reworded the caption to try to make this clearer.

Discussion is also tightly written and seems appropriate to me.

Reviewer 2 Report

I can only say that this is a nice paper, well written and organized, and I think it is perfect for this special issue. I can therefore suggest its publication in the present form.

Author Response

I can only say that this is a nice paper, well written and organized, and I think it is perfect for this special issue. I can therefore suggest its publication in the present form.

Thank you very much for your kind comments!

Reviewer 3 Report

The authors applied a Bayesian statistical model to an existing dataset from a foraging task. Compared to a previous publication, the model explains not only average statistics across trials, but target-by-target behaviour of human participants.

This is an interesting new application of the model and should be a valuable addition to the existing literature. I have a few issues regarding the presentation of the results:

Distribution of parameters: It would be interesting to see a distribution of the four parameters of the model to assess how much each of those parameters influences target selection.

Non-constant chance performance: Figure 1 illustrates very well that chance performance is not constant but depends on the number of remaining targets. Therefore, calculating the average model accuracy over all targets is somewhat misleading because it cannot be compared directly to a constant chance performance. It might be better to first calculate accuracy relative to chance performance for all target selections separately, before averaging across target selections.

Initial target selection: As Figure 8 nicely shows, there is a strong correlation between the accuracy of the prediction and the weight of the two components of the mixture model (top-left corner and center). One could argue that the prediction is only accurate when the weight for the top-left corner bias is high, meaning that the model can predict a top-left corner bias but nothing else. This casts doubts on whether the center bias is relevant at all.

Replication with second dataset: I praise the authors for checking if their results generalize to a second dataset. However, these replication results are buried in the supplementary material. It might be more useful to show both datasets in the same graphs in the paper.

Minor comments:

Figure 3: It is clear that the red arrows indicate discrepancies between human and model data, but does the shown order of targets correspond to the human’s or the model’s order?

Figure 4, right panel: I don’t quite understand the positive relationship between accuracy and the probability that the path length is shorter in the human data than in the model. Shouldn’t accuracy increase from 0 to 50% and then decrease again from 50 to 100%? To put it the other way around: how can accuracy increase if the difference in path length increases? There is very little explanation of this relationship in the text.

Author Response

The authors applied a Bayesian statistical model to an existing dataset from a foraging task. Compared to a previous publication, the model explains not only average statistics across trials, but target-by-target behaviour of human participants.

This is an interesting new application of the model and should be a valuable addition to the existing literature. I have a few issues regarding the presentation of the results:

Distribution of parameters: It would be interesting to see a distribution of the four parameters of the model to assess how much each of those parameters influences target selection.

Figure 4 in Supplementary Material 1 shows how the values of all the model parameters vary with model accuracy. Does this help? We are of course happy to produce a different figure if you have something different in mind.

Non-constant chance performance: Figure 1 illustrates very well that chance performance is not constant but depends on the number of remaining targets. Therefore, calculating the average model accuracy over all targets is somewhat misleading because it cannot be compared directly to a constant chance performance. It might be better to first calculate accuracy relative to chance performance for all target selections separately, before averaging across target selections.

We agree that the average model statistic, as with any summary stat, is imperfect. However, we have chosen to use this as it is (hopefully) more intuitive to readers than a mean of proportions. To help facilitate comparisons with the baseline, we report the average baseline accuracy over a trial (10.9%). We don’t think there is a problem with readers being misled, as the change over the course of a trial is clearly presented in Figure 1. We have also included more information about how the model accuracy statistic is calculated on L122.

Initial target selection: As Figure 8 nicely shows, there is a strong correlation between the accuracy of the prediction and the weight of the two components of the mixture model (top-left corner and center). One could argue that the prediction is only accurate when the weight for the top-left corner bias is high, meaning that the model can predict a top-left corner bias but nothing else. This casts doubts on whether the center bias is relevant at all.

For the first target selection, chance level is 1/40 = 0.025. Thus, the model is doing better than chance for most participants close to lamba = 1, suggesting that the model still has some predictive value for participants with a central bias. We have added the chance level to the graph to hopefully make this point clearer.

Replication with second dataset: I praise the authors for checking if their results generalize to a second dataset. However, these replication results are buried in the supplementary material. It might be more useful to show both datasets in the same graphs in the paper.

We have highlighted this replication more by adding a short section to the end of the results (see L190 onwards). We have avoided adding more figures to the main manuscript as a) the broad results are very similar and b) the dataset in the main text is larger, so the results should be more representative and generalisable. 

Minor comments:

Figure 3: It is clear that the red arrows indicate discrepancies between human and model data, but does the shown order of targets correspond to the human’s or the model’s order?

The numbers indicate the order the participants selected targets in. We have clarified this in the figure caption.

Figure 4, right panel: I don’t quite understand the positive relationship between accuracy and the probability that the path length is shorter in the human data than in the model. Shouldn’t accuracy increase from 0 to 50% and then decrease again from 50 to 100%? To put it the other way around: how can accuracy increase if the difference in path length increases? There is very little explanation of this relationship in the text.

The maximum accuracy is around about 0.7: therefore, our model is making errors even for the participants it is best at predicting. This graph indicates that the errors it is making for these participants are skewed i.e. the model is making longer paths than the participants themselves. For the participants that the model predicts more poorly, we see the opposite behaviour, with participants often having longer paths than the model would predict. We hope that Figure 3 makes some of these patterns clearer.